# Dysregulation of Glycerophosphocholines in the Cutaneous Lesion Caused by *Leishmania major* in Experimental Murine Models

**DOI:** 10.3390/pathogens10050593

**Published:** 2021-05-13

**Authors:** Adwaita R. Parab, Diane Thomas, Sharon Lostracco-Johnson, Jair L. Siqueira-Neto, James H. McKerrow, Pieter C. Dorrestein, Laura-Isobel McCall

**Affiliations:** 1Department of Microbiology and Plant Biology, University of Oklahoma, Norman, OK 73019, USA; adwaitaparab@ou.edu; 2Laboratories of Molecular Anthropology and Microbiome Research, University of Oklahoma, Norman, OK 73019, USA; 3Skaggs School of Pharmacy and Pharmaceutical Sciences, University of California San Diego, La Jolla, CA 92093, USA; d4thomas@health.ucsd.edu (D.T.); sharizzy@gmail.com (S.L.-J.); jlagedesiqueiraneto@health.ucsd.edu (J.L.S.-N.); jmckerrow@health.ucsd.edu (J.H.M.); pdorrestein@health.ucsd.edu (P.C.D.); 4Center for Microbiome Innovation, University of California San Diego, La Jolla, CA 92093, USA; 5Collaborative Mass Spectrometry Innovation Center, University of California San Diego, La Jolla, CA 92093, USA; 6Department of Chemistry and Biochemistry, University of Oklahoma, Norman, OK 73019, USA

**Keywords:** host metabolism, cutaneous leishmaniasis, *Leishmania major*, untargeted metabolomics, neglected tropical diseases, glycerophosphocholines

## Abstract

Cutaneous leishmaniasis (CL) is the most common disease form caused by a *Leishmania* parasite infection and considered a neglected tropical disease (NTD), affecting 700,000 to 1.2 million new cases per year in the world. *Leishmania major* is one of several different species of the *Leishmania* genus that can cause CL. Current CL treatments are limited by adverse effects and rising resistance. Studying disease metabolism at the site of infection can provide knowledge of new targets for host-targeted drug development. In this study, tissue samples were collected from mice infected in the ear or footpad with *L. major* and analyzed by untargeted liquid chromatography-tandem mass spectrometry (LC-MS/MS). Significant differences in overall metabolite profiles were noted in the ear at the site of the lesion. Interestingly, lesion-adjacent, macroscopically healthy sites also showed alterations in specific metabolites, including selected glycerophosphocholines (PCs). Host-derived PCs in the lower *m/z* range (*m/z* 200–799) showed an increase with infection in the ear at the lesion site, while those in the higher *m/z* range (*m/z* 800–899) were decreased with infection at the lesion site. Overall, our results expanded our understanding of the mechanisms of CL pathogenesis through host metabolism and may lead to new curative measures against infection with *Leishmania*.

## 1. Introduction

Leishmaniasis affects people in 88 countries worldwide in tropical, subtropical and temperate regions, putting approximately 350 million individuals at risk of infection, with approximately 12 million battling the disease [1]. It is one of the three most impactful vector-borne protozoan neglected tropical diseases, causing approximately 2.1 million DALYs (Disability-Adjusted Life Years) and 51,000 deaths annually. With recent population movements, leishmaniasis is now affecting people in non-endemic regions as well. The expanding spread of leishmaniasis can also be attributed to climate change and social constraints of populations living in poverty and conflict. Leishmaniasis is a disease that is exacerbated by poverty and socio-economic barriers, which subsequently increase the rates of disease progression, mortality and morbidity and the social stigma [2,3]. 

Leishmaniasis is caused by about 20 different species of the parasite *Leishmania*, and it is manifested most frequently in three clinical syndromes in humans: visceral, cutaneous (CL) and mucocutaneous leishmaniasis. CL is the most common form of the disease, and symptoms include skin lesions and ulcers on exposed parts of the body. Mucocutaneous leishmaniasis is a disabling form where the lesions can lead to destruction of soft tissue of the nose, mouth, and throat cavities. Of the three clinical forms of the disease, visceral leishmaniasis (kala-azar) is the deadliest, with serious symptoms such as swelling of the liver and spleen, extreme anemia and frequent bouts of fever. Infection is transmitted through female sandflies of the *Phlebotomus* genus in the Old World and the *Lutzomyia* genus in the New World. Infected female sandflies bite the mammalian host, regurgitating promastigote-stage parasites into the host’s body. Subsequently, the parasites are taken up by macrophages, neutrophils and dendritic cells. Within the macrophage phagolysosome, promastigotes differentiate into the amastigote stage, which then multiply and affect various tissue types depending on whether infection is initiated by a viscerotropic or dermotropic parasite strain [4,5]. This initiates the clinical manifestations of the disease. Humans as well as other mammals serve as host reservoirs for the parasite [6]. 

The current course of treatment for CL is usually antimonial drug compounds. These are known to be highly toxic compounds, in addition to the threat of increased parasite resistance to antimony in several regions of the world. Miltefosine, amphotericin B and paromomycin are among the other drugs that are administered for CL treatment, all of which have the drawbacks of high levels of toxicity, increased drug resistance and treatment failure. Miltefosine is also teratogenic and should not be given to women of childbearing age. Treatment failure can be attributed to the characteristics of the host (immune system and nutritional status), of the parasite (mechanisms of survival within the host, drug resistance mechanisms, tissue location, etc.) and of environmental factors such as awareness and treatment accessibility [7]. Approaching disease pathogenesis from a molecular perspective could uncover new mechanisms of infection and aid in developing new cures for leishmaniasis [8]. 

Alongside genes and proteins, metabolites play an important role in the life of an organism. The metabolome reflects the true functional endpoint of a complex biological system and provides a functional view of the organism by taking into account the sum of its genes, RNA, proteins and its environment [9]. Untargeted metabolomics can help identify metabolites involved in disease pathogenesis in an unbiased fashion, acquiring data across a broad mass range [10]. For example, untargeted metabolomics has shown that miltefosine’s mode of action *in vitro* may be related to modulation of the parasite lipid metabolism, particularly increased levels of by-products of lipid turnover [11]. Importantly, miltefosine was originally designed as a cancer treatment and thus may also target the host glycerophosphocholine (PC) metabolism [12]. PCs are important membrane constituents but also play regulatory and signaling roles, particularly pro-inflammatory [13]

The overall aim of this work was to perform an untargeted metabolic analysis of CL lesions in mice infected with *Leishmania major*. Our results showed significant changes in the host metabolism, specifically concerning PCs, in the skin lesions of CL.

## 2. Results

### 2.1. Overall Impact of L. major Infection on Ear and Footpad Metabolism

To understand better the impact of infection on tissue metabolites, we analyzed overall and specific metabolite differences in the presence and absence of infection with *Leishmania major* at sites of lesion and lesion-adjacent sites (with no visible signs of infection). BALB/c mice were injected intradermally in the ear or subcutaneously in the footpad with *L. major* parasites. Eight weeks post-infection, samples were collected from the area where the parasites were injected, which showed skin lesions (“infected ear center”), the surrounding area that appeared infection-free (“infected ear edge”), and the matched tissue regions from the uninfected ear (“uninfected ear center”, “uninfected ear edge”) (Figure 1A). Metabolites were extracted with aqueous and organic solvents and analyzed by untargeted LC-MS/MS (see Materials and Methods). Overall, for both aqueous and organic extractions, distinct global metabolite profiles were observed by Principal Coordinate Analysis (PCoA) for the infected ear center compared to the infected ear edge (PERMANOVA *p* < 0.01, aqueous extraction R^2^ = 0.743, organic extraction R^2^= 0.643), to the uninfected ear center (PERMANOVA *p* < 0.01, aqueous extraction R^2^ = 0.739, organic extraction R^2^ = 0.805) and to the uninfected ear edge (PERMANOVA *p* < 0.01, aqueous extraction R^2^ = 0.288, organic extraction R^2^ = 0.248). In contrast, no significant differences for both aqueous and organic extracts by PCoA analysis in terms of overall metabolite profile were observed between the infected ear edge and the uninfected ear samples (Figure 1B,C, PERMANOVA *p* > 0.1). Thus, *L. major* infection changes the overall tissue chemical composition at the lesion location in the ear. In contrast, the impact of *L. major* infection on the overall footpad metabolite profile for the organic (PERMANOVA *p* = 0.218 R^2^ = 0.156) and aqueous (PERMANOVA *p* = 0.244 R^2^ = 0.146) extractions was much less significant. 

Random forest machine learning analysis [14] was performed to identify the metabolites most affected by infection in both experimental systems, with annotation performed using molecular networking and GNPS [15]. Annotatable molecules most highly affected by infection include metabolites of the glycerophosphocholine family of phospholipids, including PC O-38:5 (*m/z* 794.6051 RT 4.42 min), PC O-36:3 (*m/z* 770.605 RT 4.58 min), PC O-36:4 (*m/z* 768.5862 RT 5.26 min) and PC 37:7 (*m/z* 790.5424 RT 5.42 min). Glutamine and eicosatrienoic acid were also affected by infection (Figure 2 and Appendix A, Table 1, Table 2, Table 3 and Table 4). Glutamine was decreased with infection at the site of the ear lesion (Wilcoxon rank sum test *p* value = 0.008 comparing to the uninfected ear center, Figure 1D), although it was unaffected by infection in the footpad. Eicosatrienoic acid was increased in the infected footpad (Wilcoxon rank sum test *p* value = 0.008). PE(18:1/0:0) (*m/z* 480.3097 RT 2.81 min) was also increased by infection in the footpad (Wilcoxon rank sum test *p* value = 0.008).

### 2.2. Impact of L. major Infection on Tissue PCs

Given that many of the differential molecules are PCs, we investigated the impact of infection on this family in greater detail. Molecular network analysis of the PC family molecules in both aqueous and organic ear extracts showed that most detected PCs were strongly increased by infection (Figure 3). In particular, the infected group was significantly higher than the uninfected group for PCs in the lower mass ranges of *m/z* 200–299, 400–499, 500–599 and 600–699 (Wilcoxon rank sum test *p* value < 0.05). PCs in these mass ranges were also significantly higher in the infected ear center compared to the infected ear edge, to the uninfected ear center and to the uninfected ear edge (Wilcoxon rank sum test *p* value < 0.05 for each pairwise comparison, Figure 4A–D). PC levels in the mass range of *m/z* 700–799, and total PCs were significantly increased in the infected group in comparison to the uninfected group (Wilcoxon rank sum test *p* value < 0.05, Figure 4E,G). A reverse trend was observed in the mass range of *m/z* 800–899 where the PC levels were significantly higher in the uninfected ear center in comparison to the infected ear center (Figure 4F). Given that almost all PCs were detected in both infected and uninfected samples, albeit at differential abundances, they are either host-derived or commonly produced by both parasite and host. No PCs were detected in the *m/z* 300–399 range. Likewise, most PCs were increased by infection in the footpad (Table 3 and Table 4, Figure 3). These results indicate that PCs are strongly affected by cutaneous *Leishmania* infection. In addition, our observation that specific PCs as well as PCs of multiple *m/z* ranges are also affected at lesion-adjacent sites (“infected ear edge”) indicates that infection-induced metabolic perturbations are not restricted to the lesion site, revealing a better picture of what is happening to the host during the disease state and providing clues to the pathways involved. 

## 3. Discussion

The metabolome provides a link between genotype and phenotype by identifying changes occurring at the molecular level, for example, when parasites and their hosts interact [16]. Metabolism is also an indicator of the host physiological state. Understanding the infection-induced host metabolic alterations could lead to the identification of potential targets and subsequently to drug development for parasitic diseases [17], particularly host-targeted drug therapy focused on pathways otherwise redundant to the host but important for parasite invasion, replication and survival [18], or on mitigating damage caused by the parasite [17]. In addition, changes in the host plasma metabolite abundance, including pyruvate, taurine and *N*-acetylglutamine, can serve as an indicator of response to CL treatment [19]. Several studies previously investigated *Leishmania* and host metabolism during *in vitro* macrophage infection [20,21,22,23], or in amastigotes purified from mouse granulomatous lesions [24], but there is still a lack of knowledge of host metabolic responses during *in vivo* infection. 

We cannot exclude the possibility that detected metabolites are produced by both host and parasite. However, we estimate that most infection-associated metabolites identified in our study are nevertheless host-derived for the following reasons: the relative host vs parasite biomass and our instrumental limit of detection; the slow replication of *Leishmania* during *in vivo* infection [24]; the fact that lesion-derived amastigote PC composition significantly differs from the mouse tissue PC composition [25]; most detected metabolites in our study are found in both infected and uninfected samples, including PCs (see Figure 3). Our results therefore expand our understanding of host metabolic contributions to CL pathogenesis. We note the following exceptions with regards to PCs: *m/z* 770.605 RT 4.58 min, annotated as PC O-36:3 and *m/z* 772.6201 RT 4.73 min, annotated as PC O-36:2, which were only detected in the infected ear edge and not in the uninfected samples. These findings concur with data from Moitra *et al*, which only detected these PCs in amastigotes and not in uninfected mouse tissue [25]. These findings should however be interpreted with caution, as these PCs are all listed as “endogenous” in the Human Metabolome Database (HMDB) [26,27]. Thus, they all have at least the potential to be host-derived but infection-induced. 

Amongst annotatable metabolites in our study, members of the PC family were most affected by infection in both intradermal ear infection and subcutaneous footpad infection models. PCs of the *m/z* range 200–799 and total PCs were significantly higher with infection at the site of the ear infection (infected ear center) (Figure 4). PCs were also increased in the infected footpad (Table 3 and Table 4, Figure 3). This increase concurs with prior reports of elevated lysophosphatidylcholine (LPC) and PCs in infected macrophages *in vitro* [23,28]. Concordance between infection models supports our approach and translatability of results.

LPC has immunomodulatory roles that promote parasite growth [29]. PC elevation may also reflect increased membrane turnover during infection and modulate immune responses [30]. For example, PC biosynthesis is a critical component of Golgi membrane remodeling following TLR4 engagement and is required for secretion of TNFα and interleukin 6 cytokines [31]. TLR4 is required for control of *L. major* infection [32]. Elevation of PCs is also a marker of the switch from monocyte to macrophage [33]. Increased PCs may also reflect phagolysosome membranes given the intracellular lifestyle of *L. major* [30], though PC elevation was also observed during infection with *T. cruzi,* which resides in the cytosol [34,35]. Lastly, PCs may also be derived from other immune cells attracted to the site of infection, such as T cells and dendritic cells. The absence or low levels of these cells in uninfected tissue would account for the absence of these PCs in uninfected or lesion-adjacent sites.

Miltefosine is a commonly administered oral drug for the treatment of visceral leishmaniasis and CL that targets the PC biosynthetic pathway, inhibiting phosphatidylethanolamine-N-methyltransferase and activating phospholipase A2 in *Leishmania* [36]. In mammalian cells, miltefosine also decreases phosphatidylethanolamine-N-methyltransferase activity, while also decreasing membrane-bound CTP:phosphocholine cytidylyltransferase activity, leading to reduced levels of PCs [12]. Thus, it may be expected to proceed via host-directed effects in addition to impacts on parasite metabolism. This is further supported by recent observations that PC biosynthesis is dispensable in *Leishmania* amastigotes [25]. We therefore speculate that the mechanism of action of miltefosine in CL may thus involve re-normalization of infection-induced changes in host PCs and restriction of the parasite’s ability to scavenge host PCs. Future studies are thus needed to investigate the mechanism of action of miltefosine with respect to the host metabolism in CL *in vivo*.

Additional annotatable infection-affected metabolites also included the omega-3 fatty acid eicosatrienoic acid and glutamine. Glutamine was significantly lower with infection at the site of the ear lesion but was unaffected by infection in the footpad. These findings contrast with the metabolomic profiling of *L. amazonensis*-infected macrophages, which showed increased glutamine levels [22]. A recent study in mice infected with *L. donovani,* however, showed heightened glutamine consumption with infection and a role of glutamine supplementation in clearing parasite load [37], which concur with our findings. Future studies should aim to look at the specific functional role of the glutamine metabolism in *L. major* infection. 

The clinical presentations of CL lesions can vary, and lesions are capable of self-healing in some cases. However, resolving them can take several months to years, leaving a significant amount of scarring. In cases of Post-Kala Azar dermal leishmaniasis, patients can continue to serve as a reservoir for the parasites after the lesions have long been healed [38]. Our results showed significant perturbations in the metabolism of the skin lesions, with the area near the skin lesions also being affected in experimental CL. Our study relied on bioluminescence to measure parasite burden and as such we cannot ascertain whether parasites were still present at low levels in the sites adjacent to the skin lesions. There is therefore still a strong need to understand the role of lesion-free tissues in the transmission of *Leishmania* and in disease pathogenesis. However, our results are consistent with findings of microbiota dysbiosis in lesion-adjacent tissues in humans and in lesion-free cutaneous sites in mice [39].

This study looked at both ear and footpad infection models, although the effect of infection on metabolism in the footpad was found to be less significant than in the ear. Nevertheless, PC family metabolites were increased with infection in both sites, showcasing similarities in pathogenesis processes between these two infection models. These similarities are particularly striking given differences in pathogenic processes between ear and footpad models, including differences in the elicited immune response [40,41] and vaccine-mediated protection [42]. These differences in immune cell recruitment may account for the observed differences in the infection-elicited PC profile.

While this untargeted metabolomics study enabled us to uncover several metabolic pathways affected in CL, on average, compounds that could not be directly annotated (level 2 annotations according to metabolites standards initiative [43]) still represent 71.11% of our data. Molecular networking did enable us to extend annotations further, so that 41.6% of our top 15 most differential metabolite features identified by random forest had at least family-level (level 3) annotations [43]. Nevertheless, metabolomics annotation rates are continuously improving. Our results were deposited in a “living data” database [15], where they are continuously being re-annotated as reference libraries and computational tools expand. As such, they will continue to yield expanding insights into CL pathogenesis and serve as a building point for expanded studies of metabolism in CL. Such results will help guide the next generation of CL drug treatments. 

## 4. Materials and Methods

### 4.1. In Vivo Experimentation

Female BALB/c mice (6–8 week-old) were injected intradermally in the left ear with 1 × 10^6^ luciferase-expressing *L. major* strain LV39 promastigotes (n = 5) or in the left rear footpad with 5 × 10^6^ luciferase-expressing *L. major* strain LV39 promastigotes in PBS (n = 5) [44]. These inocula were selected based on standard inocula for each model [45,46]. Bioluminescence imaging was performed on an IVIS Lumina LT Series III instrument (Perkin Elmer), and data were processed using Living Image 4.5 software (Perkin Elmer, https://www.perkinelmer.com/product/li-software-for-spectrum-1-seat-add-on-128113, used September 2016–May 2021). Upon euthanasia, infected and uninfected ear tissue, including the entirety of the lesion area (“infected ear center”) and the entirety of the surrounding, macroscopically-healthy surrounding area (“infected ear edge”), as well as matched positions from the other, uninfected ear (“uninfected ear center”, “uninfected ear edge”) were collected 8 weeks post-infection and immediately snap-frozen. Upon euthanasia, infected and uninfected footpads were collected 7 weeks post-infection; lesion tissue was scraped off above the footpad bones and collected in its entirety, with matched tissue collected from the other, uninfected footpad. The entirety of the tissue was immediately snap-frozen. Samples were stored at −80 °C until metabolite extraction. Parasites were maintained at 28 °C in M199 medium (Sigma) supplemented with 10% fetal bovine serum (Sigma), 1% penicillin-streptomycin, RPMI 1640 vitamin mix (1%), HEPES (25 mM), adenosine (100 μM), glutamine (1 mM), hemin (0.005%), NaHCO_3_ (12 mM) and folic acid (10 μM) (pH 7.2) [47].

### 4.2. LC-MS/MS

Metabolite extraction, liquid chromatography and mass spectrometry were performed as previously described [48]. Briefly, metabolites were extracted by homogenization in a Tissuelyzer using a 5 mm stainless steel bead (Qiagen) in 50% methanol (aqueous extract) followed by 3:1 dichloromethane:methanol (organic extract). LC was performed on an UltiMate 3000 UHPLC (Thermo Scientific, Waltham, MA, USA) with Phenomenex UHPLC 1.7 µm 100 Å Kinetex C8 column (50 × 2.1 mm), and with water and 0.1% formic acid as mobile phase A and acetonitrile and 0.1% formic acid as mobile phase B, flow rate of 0.5 mL/min and column temperature of 40 °C. LC gradient parameters were optimized for each extraction with regards to overall chromatogram peak shape (Table 5). Daily MS calibration was performed with the ESI-L Low Concentration Tuning Mix, which covers *m/z* 118.086 to 2721.895 (Agilent Technologies). The internal calibrant (lock mass) Hexakis(1H,1H,3H-tetrafluoropropoxy)phosphazene (Synquest Laboratories), *m/z* 922.009798, was present throughout the run, as previously described [48,49,50]. MS was performed in the positive mode on a Maxis Impact HD QTOF mass spectrometer (Bruker); instrumental parameters are listed in Table 6. MS/MS data for each run was collected by fragmentation of the ten most intense ions, in a range of 80–2000 *m/z*, with active exclusion after 4 spectra and release after 30 s. Instrumental performance controls included solvent blanks, pooled quality controls for each tissue type and a standard mix of 6 molecules (sulfamethazine, sulfadimethoxine, sulfachloropyridazine, coumarin-314, sulfamethizole, amitriptyline). To avoid any confounding from run order, we alternated between samples from infected and uninfected animals.

### 4.3. LC-MS/MS Data Analysis

LC-MS/MS data were processed using MZmine 2.37 [51], with parameters as shown in Table 7. 

Features with a peak area within 3-fold of peak area in blanks were removed. Normalization to total peak area (Total ion current (TIC) normalization) and data processing were performed in Jupyter notebook in R [52]. Principal Coordinate Analysis (PCoA) was done using the Bray-Curtis dissimilarity matrix implemented in QIIME1 [53], and PERMANOVA calculations were performed using the R package “vegan” to compare the chemical similarity of samples from the four groups of varying condition and position of infection [54,55]. EMPeror was used to visualize PCoA plots [56]. The randomForest package in R was used to find variables of importance associated with infection and sampling conditions, using 7000 trees [14]. The Global Natural Products Social Molecular Networking platform (GNPS) was used to annotate molecules from spectral library references and to perform feature-based molecular networking [15,57,58]. The following parameters were used in GNPS: precursor ion mass tolerance of 0.02 Da, fragment ion mass tolerance of 0.02 Da, minimum cosine score of 0.7 and 4 or more matched fragment ions. The maximum shift allowed between two MS/MS spectra was 500 Da, 10 maximum neighbor nodes allowed and a maximum difference between the precursor ion mass of searched MS/MS spectrum and library spectra was 100 Da. Spectral matches were evaluated by considering cosine scores, quality of mirror plots, as well as the number of matched peaks. Molecular network visualization was done in Cytoscape 3.7.2 [59]. All members of the PC subnetworks were visually inspected and verified to contain the diagnostic MS/MS peaks with *m/z* 184.08 (phosphocholine), *m/z* 125.00 (2,2-Dihydroxy-1,3,2-dioxaphospholan-2-ium) and *m/z* 86.10 (N,N,N-Trimethylethenaminium) from the phospholipid head group. Putative annotations for members of the PC subnetworks that were not available through spectral matching in GNPS were obtained using LipidMaps [60]. Notched box plots showing metabolite feature abundance for the four different groups (infected/uninfected vs. center/edge) for the ear samples and two different groups (infected vs. uninfected) for the footpad samples along with non-parametric two-tailed Wilcoxon statistical tests were both performed in R. Boxplot whiskers represent the lowest and largest data points, and non-overlapping boxplot notches indicate different medians between groups (95% confidence) [61]. Molecular formula prediction was performed using SIRIUS version 4.5.1 [62], allowing 30 ppm mass deviation, with 10 candidates, searching all available databases and considering [M+H]^+^, [M+K]^+^ and [M+Na]^+^ ions. Allowed elements were: C, H, N, O, P, S. The formula with the highest Sirius Score is presented here.

## Figures and Tables

**Figure 1 pathogens-10-00593-f001:**
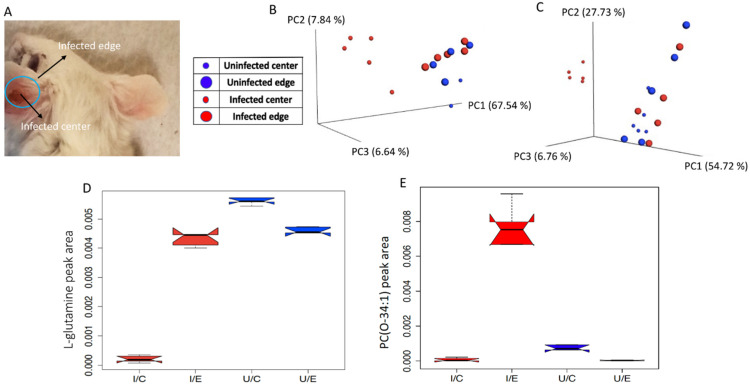
Effect of *in vivo L. major* infection on host metabolite profile. (**A**) Sites of infection and sample collection. Lesion at the center of the infected ear is circled in blue. (**B**) PCoA analysis of aqueous extraction from infected and uninfected ear samples, showing overall differences in metabolite profiles between sampling sites: PERMANOVA *p* = 0.004, R^2^ = 0.288. (**C**) PCoA analysis of organic extraction from infected and uninfected ears, showing differences in global metabolite profiles between sampling sites: PERMANOVA *p* = 0.003, R^2^ = 0.248. (**D**) Representative metabolite decreased by infection at the site of the lesion: glutamine (Wilcoxon rank-sum test comparing infected ear center vs infected ear edge *p* = 0.008). (**E**) Representative metabolite increased only at infection-adjacent sites: PC(O-34:1), Wilcoxon rank-sum test comparing infected ear center vs infected ear edge *p* = 0.008. Non-overlapping boxplot notches indicate significantly different medians between groups.

**Figure 2 pathogens-10-00593-f002:**
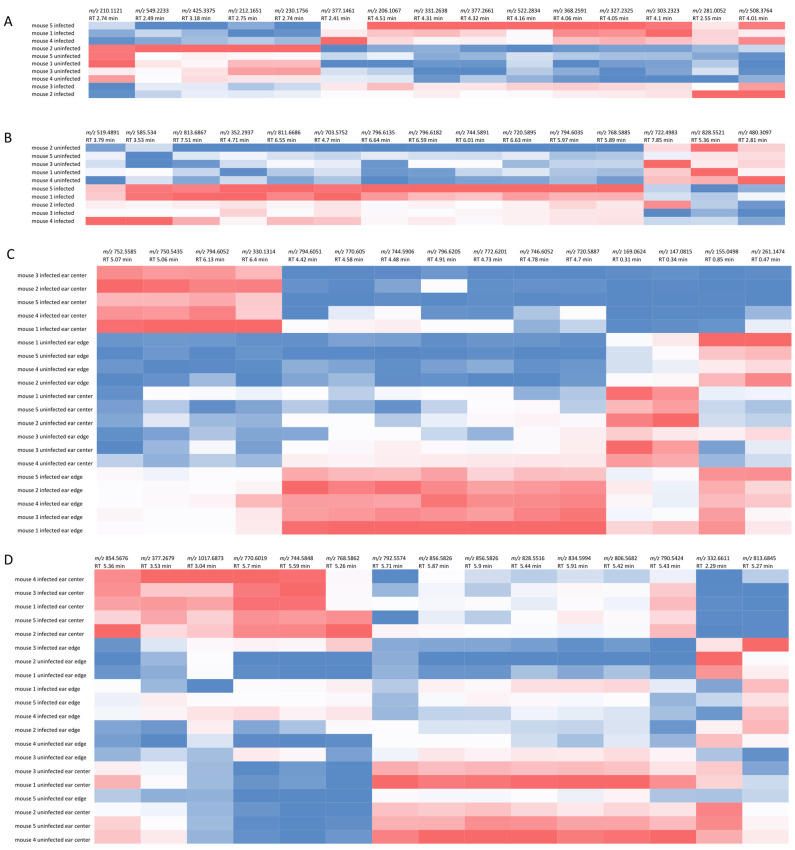
Top 15 infection-modulated metabolites as identified by random forest. (**A**) Footpad aqueous extraction. (**B**) Footpad organic extraction. (**C**) Ear aqueous extraction. (**D**) Ear organic extraction.

**Figure 3 pathogens-10-00593-f003:**
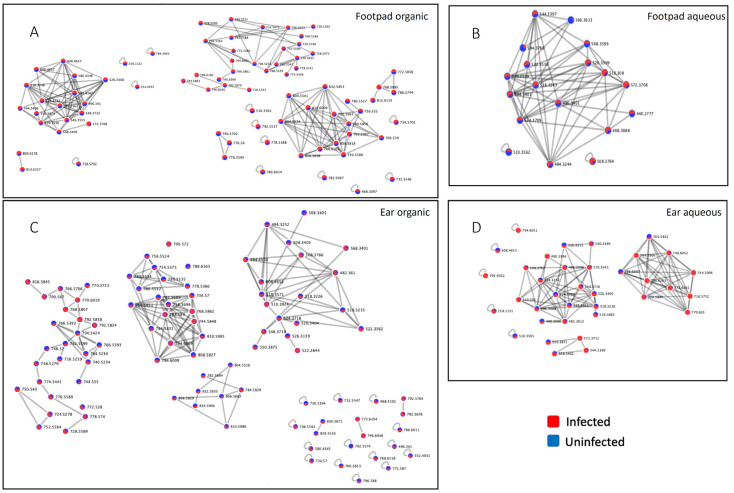
PC subnetworks. PC family metabolites in the footpad organic extraction (**A**), footpad aqueous extraction (**B**), ear organic extraction (**C**) and ear aqueous extraction (**D**) molecular networks. Relative metabolite abundance in the presence of infection is shown in red, and absence of infection is blue.

**Figure 4 pathogens-10-00593-f004:**
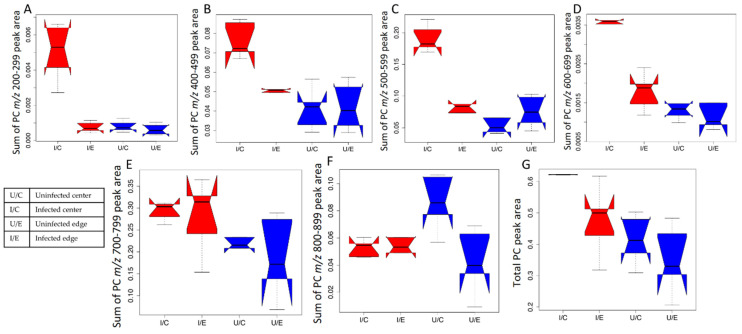
(**A**–**F**) PCs in the *m/z* range 200–299, 400–499, 500–599, 600–699, 700–799 and 800–899, respectively, change with infection and sampling position in the ear. (**G**) Total PC levels were increased at the site of infection in the ear. Non-overlapping boxplot notches indicate significantly different medians between groups.

**Table 1 pathogens-10-00593-t001:** Top differential molecules for ear aqueous extraction as determined by random forest.

*m/z*	RT (min)	Predicted Molecular Formula (SIRIUS) or Spectral Match Formula	Spectral Match on GNPS/LIPID MAPS/Molecular Networking	Mass Difference	PPM Error	Cosine Score	Number of Matched Peaks for GNPS Spectral Match	*p* Value (Infected Center vs Uninfected Center) ^1^	*p* Value (Infected Edge vs Uninfected Edge)	*p* Values (Infected vs Uninfected)
Dysregulated in ear center										
147.0815	0.34	C_5_H_10_N_2_O_3_	Glutamine	0.01	31	1	4	0.008	0.421	0.001
169.0624	0.31	C_4_H_12_N_2_O_3_S	NA ^2^	NA	NA	NA	NA	0.011	0.548	0.011
330.1314	6.4	C_14_H_21_N_5_O_2_ ([M+K]^+^)	NA	NA	NA	NA	NA	0.008	0.008	1.08 × 10^−5^
**750.5435 ^3^**	**5.06**	**C_34_H_17_N_9_O_9_**	**NA**	**NA**	**NA**	**NA**	**NA**	**0.008**	**0.008**	2.17 × 10^−5^
**752.5585**	**5.07**	**C_40_H_73_N_5_O_8_**	**NA**	**NA**	**NA**	**NA**	**NA**	**0.008**	**0.008**	1.08 × 10^−5^
**794.6052**	**6.13**	**C_46_H_84_NO_7_P**	**PC O-38:5**	**0**	**0**	**1**	**5**	**0.008**	**0.008**	1.08 × 10^−5^
Dysregulated in ear edge										
261.1474	0.47	C_9_H_24_N_3_O_2_S ([M+Na]^+^)	NA	NA	NA	NA	NA	0.095	0.31	0.315
720.5887	4.7	C_40_H_82_NO_7_P	LPC 32:0 or LPC O-32:1;O or PC O-32:0	NA	2	NA	NA	0.095	0.008	0.28
744.5906	4.48	C_42_H_82_NO_7_P	LPC 34:2 or LPC O-34:3;O or PC O-34:2	NA	0.54	NA	NA	0.008	0.008	0.052
**746.6052**	**4.78**	**C_42_H_84_NO_7_P**	**PC O-16:0/18:1**	**0**	**4**	**1**	**14**	**0.032**	**0.008**	**0.28**
770.605	4.58	C_44_H_84_NO_7_P	PC O-36:3	NA	1.04	NA	NA	0.310	0.008	0.218
772.6201	4.73	C_44_H_86_NO_7_P	PC O-36:2	NA	1.8	NA	NA	0.016	0.008	0.393
**794.6051**	**4.42**	**C_46_H_84_NO_8_P**	**PC O-38:5**	**0**	**0**	**1**	**5**	**0.056**	**0.008**	**0.481**
**796.6205**	**4.91**	**C_46_H_86_NO_7_P**	**PC O-38:4**	**NA**	**1.26**	**NA**	**NA**	**0.151**	**0.008**	**0.105**
Dysregulated in both ear center and ear edge										
155.0498	0.85	C_11_H_6_O	NA	NA	NA	NA	NA	0.008	0.222	0.684

^1^ Test returns identical *p*-values when non-overlapping peak areas are obtained between samples in each group (due to identical ranks regardless of the specific peak areas);^2^ NA, not applicable (no spectral match in GNPS); ^3^ Bolding indicates that metabolite is also affected by infection in the footpad.

**Table 2 pathogens-10-00593-t002:** Top differential molecules for ear organic extraction as determined by random forest.

*m/z*	RT (min)	Predicted Molecular Formula (SIRIUS) or Spectral Match Formula	Spectral Match on GNPS	Mass Difference	PPM Error	Cosine Score	Number of Matched Peaks for GNPS Spectral Match	*p* Values (Infected Center vs Uninfected Center) ^1^	*p* Values (Infected Edge vs Uninfected Edge)	*p* Values (Infected vs Uninfected)
Dysregulated in ear center										
377.2679	3.53	C_11_H_34_N_10_O_3_ ([M+Na]^+^)	NA ^2^	NA	NA	NA	NA	0.008	0.222	0.043
744.5848	5.59	C_42_H_82_NO_7_P	LPC 34:2 or LPC O-34:3;O or PC O-34:2	NA	7.25	NA	NA	0.011	0.052	0.0003
**768.5862 ^3^**	**5.26**	**C_44_H_82_NO_7_P**	**PC O-36:4**	**NA**	**5.2**	**NA**	**NA**	**0.008**	**0.008**	**1.08 × 10^−5^**
770.6019	5.7	C_44_H_84_NO_7_P	PC O-36:3	NA	5.06	NA	NA	0.008	0.095	0.0001
790.5424	5.43	C_45_H_76_NO_8_P	PC 37:7	NA	5.44	NA	NA	0.548	0.691	0.631
792.5574	5.71	C_45_H_78_NO_8_P	PC O-16:0/22:6	0.01	8	0.86	7	0.012	0.691	0.026
806.5682	5.42	C4_6_H_80_NO_8_P	PC 38:6 or PC O-38:7;O	0.02	22	0.81	18	0.008	1	0.09
828.5516	5.44	C_48_H_78_NO_8_P	PC O-40:10;O or PC 40:9	NA	1.2	NA	NA	0.008	0.691	0.143
834.5994	5.91	C_48_H_84_NO_8_P	PC 40:6 or PC O-40:7;O	NA	1.56	NA	NA	0.008	0.548	0.353
854.5676	5.36	C_38_H_79_N_9_O_10_S	NA	NA	NA	NA	NA	0.056	0.222	0.218
856.5826	5.87	C_43_H_73_N_11_O_7_	NA	NA	NA	NA	NA	0.008	1	0.105
856.5826	5.9	C_43_H_73_N_11_O_7_	NA	NA	NA	NA	NA	0.008	0.841	0.075
1017.687	3.04	C_45_H_84_N_20_O_7_	NA	NA	NA	NA	NA	0.008	0.151	0.002
Dysregulated in ear edge										
**813.6845**	**5.27**	**C_46_H_92_N_4_O_5_S**	**NA**	**NA**	**NA**	**NA**	**NA**	**0.016**	**0.008**	**0.912**
Dysregulated in both ear center and ear edge										
332.6611	2.29	no prediction in SIRIUS	NA	NA	NA	NA	NA	0.008	0.222	0.0003

^1^ Test returns identical *p*-values when non-overlapping peak areas are obtained between samples in each group (due to identical ranks regardless of the specific peak areas); ^2^ NA, not applicable (no spectral match in GNPS); ^3^ Bolding indicates that metabolite is also affected by infection in the footpad.

**Table 3 pathogens-10-00593-t003:** Top differential molecules for footpad aqueous extraction as determined by random forest.

*m/z*	RT (min)	Predicted Molecular Formula (SIRIUS) or Spectral Match Formula	Spectral Match on GNPS/LIPID MAPS/Molecular Networking	Mass Difference	PPM Error	Cosine Score	Number of Matched Peaks for GNPS Spectral Match	*p* Values ^1^
206.1067	4.51	C_7_H_16_N_3_O_2_P	NA ^2^	NA	NA	NA	NA	0.008
210.1121	2.74	C_12_H_15_N_2_ ([M+Na]^+^)	NA	NA	NA	NA	NA	0.008
212.1651	2.75	C_12_H_21_NO_2_	NA	NA	NA	NA	NA	0.008
230.1756	2.74	C_12_H_23_NO_3_	NA	NA	NA	NA	NA	0.008
281.0052	2.55	C_9_H_13_O_4_PS_2_	NA	NA	NA	NA	NA	0.008
**303.2312**	**4.1**	**C_20_H_32_O_3_ ([M+H-H_2_O]^+^)**	**5,6-Epoxy-8Z,11Z,14Z-eicosatrienoic acid**	**0**	**4**	**0.89**	**8**	**0.008 ^3^**
327.2325	4.05	**C_22_H_30_O_2_**	NA	NA	NA	NA	NA	0.008
331.2638	4.31	C_22_H_34_O_2_	NA	NA	NA	NA	NA	0.008
368.2591	4.06	C_24_H_33_NO_2_	NA	NA	NA	NA	NA	0.008
377.1461	2.41	C_14_H_24_N_4_O_6_S	NA	NA	NA	NA	NA	0.012
**377.2661**	**4.32**	**C_18_H_36_N_2_O_6_**	**NA**	**NA**	**NA**	**NA**	**NA**	**0.008**
425.3375	3.18	C_24_H_44_N_2_O_4_	NA	NA	NA	NA	NA	0.008
**508.3764**	**4.01**	**C_26_H_54_NO_6_P**	**PC(P-18:0/0:0)**	**0**	**2**	**0.91**	**10**	**0.008**
**522.2834**	**4.16**	**C_24_H_44_NO_9_P**	**NA**	**NA**	**NA**	**NA**	**NA**	**0.008**
549.2233	2.49	C_22_H_36_N_4_O_10_S	NA	NA	NA	NA	NA	0.008

^1^ Test returns identical *p*-values when non-overlapping peak areas are obtained between samples in each group (due to identical ranks regardless of the specific peak areas.); ^2^ NA, not applicable (no spectral match in GNPS); ^3^ Bolding indicates that metabolite is also affected by infection in the ear.

**Table 4 pathogens-10-00593-t004:** Top differential molecules for footpad organic extraction as determined by random forest.

*m/z*	RT (min)	Predicted Molecular Formula (SIRIUS) or Spectral Match Formula	Spectral Match on GNPS/LIPID MAPS/Molecular Networking	Mass Difference	PPM Error	Cosine Score	Number of Matched Peaks for GNPS Spectral Match	*p* Values ^1^
**352.2937**	**4.71**	**C_16_H_40_N_4_O_2_P**	**NA ^2^**	**NA**	**NA**	**NA**	**NA**	**0.008 ^3^**
**480.3097**	**2.81**	**C_23_H_46_NO_7_P**	**PE(18:1/0:0)**	**NA**	**2**	**NA**	**NA**	**0.008**
519.4891	3.79	C_29_H_65_N_3_O_2_P	NA	NA	NA	NA	NA	0.008
**585.534**	**3.53**	**C_33_H_68_N_4_O_4_**	**NA**	**NA**	**NA**	**NA**	**NA**	**0.016**
**703.5752**	**4.7**	**C_40_H_75_N_6_O_2_P**	**NA**	**NA**	**NA**	**NA**	**NA**	**0.008**
**720.5895**	**6.63**	**C_44_H_79_N_3_OS ([M+Na]^+^)**	**NA**	**NA**	**NA**	**NA**	**NA**	**0.008**
722.4983	7.85	C_42_H_69_NO_7_ ([M+Na]^+^)	NA	NA	NA	NA	NA	0.095
**744.5891**	**6.01**	**C_42_H_82_NO_7_P**	**LPC 34:2 or LPC O-34:3;O or PC O-34:2**	**NA**	**2**	**NA**	**NA**	**0.008**
**768.5885**	**5.89**	**C_44_H_82_NO_7_P**	**PC O-36:4**	**NA**	**2.21**	**NA**	**NA**	**0.008**
**794.6035**	**5.97**	**C_46_H_84_NO_7_P**	**PC O-38:5**	**0**	**3**	**0.81**	**7**	**0.008**
**796.6135**	**6.64**	**C_44_H_83_N_7_O_3_ ([M+K]^+^)**	**NA**	**NA**	**NA**	**NA**	**NA**	**0.008**
**796.6182**	**6.59**	**C_46_H_86_NO_7_P**	**PC O-38:4**	**NA**	**4**	**NA**	**NA**	**0.008**
**811.6686**	**6.55**	**C_47_H_91_N_2_O_6_P**	**SM 42:3;O2**	**NA**	**0**	**NA**	**NA**	**0.008**
**813.6867**	**7.51**	**C_47_H_93_N_2_O_6_P**	**SM 18:1;O2/24:1**	**0**	**4**	**0.91**	**6**	**0.008**
**828.5521**	**5.36**	**C_37_H_78_N_7_O_11_P**	**NA**	**NA**	**NA**	**NA**	**NA**	**0.008**

^1^ Test returns identical *p*-values when non-overlapping peak areas are obtained between samples in each group (due to identical ranks regardless of the specific peak areas.); ^2^ NA, not applicable (no spectral match in GNPS); ^3^ Bolding indicates that metabolite is also affected by infection in the ear.

**Table 5 pathogens-10-00593-t005:** LC gradient parameters.

**Ear Aqueous Extraction**	
Start	2% B
1 min	2% B
1.5 min	40% B
4 min	98% B
5 min	98% B
6 min	2% B
7 min	2% B
**Ear Organic Extraction**	
Start	2% B
1 min	2% B
1.5 min	60% B
5.5 min	98% B
7.5 min	98% B
8.5 min	2% B
10.5 min	2% B
**Footpad Aqueous Extraction**	
Start	2% B
1 min	2% B
1.5 min	40% B
6 min	98% B
6.5 min	98% B
7 min	2% B
**Footpad Organic Extraction**	
Start	2% B
1 min	2% B
1.5 min	70% B
7 min	98% B
8 min	98% B
9 min	2% B
10.5 min	2% B

**Table 6 pathogens-10-00593-t006:** MS parameters.

Detection Mode	Positive
Nebulizer gas pressure	2 Bar
Capillary voltage	4500 V
Ion source temperature	200 °C
Dry gas flow	9.0 L/min
Spectra rate acquisition	3 spectra/s

**Table 7 pathogens-10-00593-t007:** MZmine parameters.

**Mass Detection**	
MS level 1: Noise level	1 × 10^3^
MS level 2: Noise level	10
Mass detector	Centroid
**Chromatogram Builder**	
Min time span	0.06 min
Min peak height	3 × 10^3^
*m/z* tolerance	1 × 10^−6^ or 10 ppm
**Chromatogram Deconvolution**	
Algorithm	Baseline cutoff
Min peak height	3 × 10^3^
Peak duration range (min)	0.06–2 min (ear), 0.01–7 min (footpad)
Baseline level	1 × 10^2^ (ear), 1.5 × 10^3^ (footpad)
*m/z* range for MS2 scan pairing (Da)	0.01
RT range for MS2 scan pairing (min)	0.2 min
**Isotopic Peaks Grouper**	
*m/z* tolerance	1 × 10^−6^ or 10 ppm
Retention time tolerance (absolute: min)	0.05 min
Monotonic shape	Enabled
Maximum charge	3
Representative isotope	Most intense
**Join Aligner**	
*m/z* tolerance	1 × 10^−6^ or 10 ppm
Weight for *m/z*	7
Retention time tolerance (absolute: min)	0.5 min
Weight for RT	3
**Manual Filtering**	
Min number of peaks per row	3
RT range	0.2–10.5 (ear organic and footpad), 0.2–6.9 (ear aqueous)
MS2	required
Manual validation of peak shape	
**Gap-Filing**	
*m/z* tolerance	1 × 10^−6^ or 10 ppm
RT tolerance	0.5 min
Intensity tolerance	30%
RT correction	Enabled

## Data Availability

Data were deposited in MassIVE (massive.ucsd.edu, accession numbers MSV000081004 (ear) and MSV000080239 (footpad)). Molecular networking can be accessed here: https://gnps.ucsd.edu/ProteoSAFe/status.jsp?task=451754c383de461e9e4abdf6eb3199d2 (aqueous ear extraction), https://gnps.ucsd.edu/ProteoSAFe/status.jsp?task=0d092bbb213347c3bd7a19b9cae2bcf4 (organic ear extraction), https://gnps.ucsd.edu/ProteoSAFe/status.jsp?task=eccdfd5f15be491b8993084c010467ef (aqueous footpad extraction), https://gnps.ucsd.edu/ProteoSAFe/status.jsp?task=ecc0d126d74f4e6e8e3d599296d8f6e2 (organic footpad extraction).

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
