# Peer review of "Dysregulation of Glycerophosphocholines in the Cutaneous Lesion Caused by Leishmania major in Experimental Murine Models"

_pathogens, 2021, doi:10.3390/pathogens10050593_

Round 1

Reviewer 1 Report

Suggestions to improve the manuscript:

The authors should inform the number of animals per group. A single animal per group is not acceptable, unless justified and discussed in the manuscript.

Tables 1 to 4 should be compacted, to fit a table per page. For this, use abbreviations to reduce the length of some periods. Tables 5 and 7 also would be better visualized without breaks between pages.

Author Response

1. The authors should inform the number of animals per group. A single animal per group is not acceptable, unless justified and discussed in the manuscript.

Response: n=5 animals per group were included. This information can be found in Materials and Methods, line 307-308 simple markup view, lines 459-461 full markup view.

2. Tables 1 to 4 should be compacted, to fit a table per page. For this, use abbreviations to reduce the length of some periods. Tables 5 and 7 also would be better visualized without breaks between pages.

Response: As recommended, we have condensed the information in Tables 1 to 4 and arranged page breaks so that Table 5 is not split between pages. Due to extra lines requested in Table 7 by reviewer 3, Table 7 cannot fit into a single page in Word. Nevertheless, formatting will improve once this paper is published in MDPI format.

Reviewer 2 Report

In this research article, Parab et al. studied the in vivo metabolic profile in L.  major infected BALB/c ear and footpad tissuesThe results of the study shed some light in the host-metabolism during CL pathogenesis while their derived  database could serve as a source of more information on host metabolism's role in CL pathogenesis upon future metabolite annotations.

The results section is short and could be slighly expanded in order to present more extensively and clearer the results derived form the study for the convenience of the reader.

Authors should also consider the following minor indications.

Abstract: 

Lines 23-24: Please change the sentence. Maybe  "...can lead to new drug targets" can be replaced with "....can provide knowledge of new targets for host-targeted drug development". 

Line 28: Change select with selected "...including selected glycerophosphocholines..."

Introduction:

Line 43: Add the word also. "...of leishmaniasis can also be attributed...."

Line 44: change the sentence to be more understanting for the reader. e.g: "....barriers which subsequently increase the rates.....and the social stigma"

Lines 47-48: "....parasite Leishmania and it is manifested most frequently in three clinical syndomes..."

Line 55: Please add "the host's body"

Line 56: Change the sentence "Subsequently, the parasites are taken up...."

Line 80: Add the word concerning "host metabolism, specifically concerning glycerophosphocholines (PC)..." 

Results:

Line 86: Add "....intradermally in the ear with L. major parasites"

Please refer the infection in the footpad as well.

Lines 82-110: Please expand this  section and provide more details of the results.

Lines 103: change "much more minor' to "much less significant"

Figure 1: Please consider enlarging the Fig1.D or provide it in a separate figure to be more clear to the reader.

Figure S2: Please consider moving the figure from the supplementary material in the article so that the differences between footpad and ear metabolite profile to enrich the information in the result section.

Discussion:

Line 181: Please change the sentence ".........alterations could lead to the identification of potential targets and subsequently to drug development [16], particularly..."

Line 187: delete the e.g., and perhaps add more reference of studies of metabolism in in vitro macrophage infection. 

Line 191: add the before mouse "..differs from the mouse tissue"

Lines189-194: Please find and add studies on Leishmania metabolites to compare with your data and enforce your conclusion that most metabolites are host-derived.

Line 198: " ...at the site of ear infection.."

Line 227: Please consider to compare with the glutamine levels in the footpad lesion.

Line 245: "....on metabolism in the footpad was found to be less significant than that of the ear".

Materials and methods:

Line 268: delete the brackets and the words "see" and "and" between thw references

Table 7-chromatogram deconvolution: Is it possible to have the format changed to have one line per data?

Table 7-Manual filtering: there is an empty row after RT range that can be deleted

Line 330: Delete the one space gap before the "Putative annotations..."

References: Please revise and use italic for genus or species (Leishmania, infantum, donovani, mexicana, major, amazonensis, T. cruzi etc) and the word De novo. In line 485 add a space gap between words "major" and "and".

Author Response

1. The results section is short and could be slightly expanded in order to present more extensively and clearer the results derived from the study for the convenience of the reader.

Response: As recommended, we have added additional details to the results section in the text (lines 123-189 in simple markup view, lines 115-213 in full markup view), as well as additional figures (Figures 2 and 3).

2. Authors should also consider the following minor indications.

Abstract: 

Lines 23-24: Please change the sentence. Maybe  "...can lead to new drug targets" can be replaced with "....can provide knowledge of new targets for host-targeted drug development". 

Response: we have made the recommended change (line 23-24 in simple markup view, lines 28-29 in full markup view).

3. Line 28: Change select with selected "...including selected glycerophosphocholines..."

Response: we have made the recommended change (line 29 in simple markup view, line 33 in full markup view).

4. Introduction:

Line 43: Add the word also. "...of leishmaniasis can also be attributed...."

Response: we have made the recommended change (line 43 in simple markup view, line 48 in full markup view).

5. Line 44: change the sentence to be more understanting for the reader. e.g: "....barriers which subsequently increase the rates.....and the social stigma"

Response: we have made the recommended change (lines 45-46 in simple markup view, lines 50-51 in full markup view).

6. Lines 47-48: "....parasite Leishmania and it is manifested most frequently in three clinical syndomes..."

Response: we have made the recommended change (lines 47-48 in simple markup view, lines 52-53 in full markup view).

7. Line 55: Please add "the host's body"

Response: we have made the recommended change (line 56 in simple markup view, line 62 in full markup view).

8. Line 56: Change the sentence "Subsequently, the parasites are taken up...."

Response: we have made the recommended change (lines 57 in simple markup view, line 63-65 in full markup view).

9. Line 80: Add the word concerning "host metabolism, specifically concerning glycerophosphocholines (PC)..." 

Response: we have made the recommended change (lines 87 in simple markup view, line 93 in full markup view).

10. Results:

Line 86: Add "....intradermally in the ear with L. major parasites"

Response: we have made the recommended change (lines 93 in simple markup view, line 100 in full markup view).

11. Please refer the infection in the footpad as well.

Response: we have made the recommended change (lines 93-94 in simple markup view, lines 100-101 in full markup view). Sentence now reads “BALB/c mice were injected intradermally in the ear or subcutaneously in the footpad with L. major parasites.”. 

12. Lines 82-110: Please expand this  section and provide more details of the results.

Response: As recommended, we have added additional details (lines 123-134 in simple markup view and lines 138-149 in full markup view).

13. Lines 103: change "much more minor' to "much less significant"

Response: we have made the recommended change (line 110 in simple markup view, lines 117-118 in full markup view).

14. Figure 1: Please consider enlarging the Fig1.D or provide it in a separate figure to be more clear to the reader.

Response: Fig1D information is now provided as part of the larger Figure 3, as recommended.

15. Figure S2: Please consider moving the figure from the supplementary material in the article so that the differences between footpad and ear metabolite profile to enrich the information in the result section.

 Response: As recommended, we have moved this figure to the main manuscript (now Figure 3).

16. Discussion:

Line 181: Please change the sentence ".........alterations could lead to the identification of potential targets and subsequently to drug development [16], particularly..."

Response: we have made the recommended change (lines 205-207 in simple markup view, lines 231-232 in full markup view).

17. Line 187: delete the e.g., and perhaps add more reference of studies of metabolism in in vitro macrophage infection. 

Response: we have made the recommended change (line 213 in simple markup view, line 238 in full markup view).

18. Line 191: add the before mouse "..differs from the mouse tissue"

Response: we have made the recommended change (line 221 in simple markup view; line 246 in full markup view).

19. Lines189-194: Please find and add studies on Leishmania metabolites to compare with your data and enforce your conclusion that most metabolites are host-derived.

Response: As recommended, we have added additional references and strengthened our rationale of the host origin of infection-perturbed metabolites (lines 216-231 in simple markup view, lines 241-255 in full markup view).

20. Line 198: " ...at the site of ear infection.."

Response: we have made the recommended change (lines 235 in simple markup view; line 314 in full markup view).

21. Line 227: Please consider to compare with the glutamine levels in the footpad lesion.

Response: As recommended, we have performed this comparison. Glutamine was not affected by infection in the footpad. We have added this information to the results section and to the discussion section (lines 131 in simple markup view and line 146 in full markup view; lines 265-266 in simple markup view and line 291 in full markup view).

22. Line 245: "....on metabolism in the footpad was found to be less significant than that of the ear".

Response: we have made the recommended change (lines 286 in simple markup view and line 311 in full markup view).

23. Materials and methods:

Line 268: delete the brackets and the words "see" and "and" between thw references

Response: we have made the recommended change (line 310 in simple markup view and line 336 in full markup view).

24. Table 7-chromatogram deconvolution: Is it possible to have the format changed to have one line per data?

Response: we have made the recommended change (line 354 in simple markup view and line 380 in full markup view).

25. Table 7-Manual filtering: there is an empty row after RT range that can be deleted

Response: we have made the recommended change (lines 354 in simple markup view and line 380 in ful markup view).

26. Line 330: Delete the one space gap before the "Putative annotations..."

Response: we have made the recommended change (lines 376 in simple markup view and line 402 in full markup view).

27. References: Please revise and use italic for genus or species (Leishmania, infantum, donovani, mexicana, major, amazonensis, T. cruzi etc) and the word De novo. In line 485 add a space gap between words "major" and "and".

Response: we have made the recommended change throughout the reference section.

Reviewer 3 Report

The authors identified a modualtion in the PC content into skin lesiona promoter by L. major infection, using the untarget metabolomic profile analysis by LC-MS/MS. I recommend the major revision of the manuscript, improviso the data description and presentation to highlights the importante finds to readers.

Title: I suggest to change the title "Dysregulation of Glycerophosphocholines in the experimental cutaneous lesion by Leishmania major in murine model".

Abstract: I suggest to revise the first phrase "Each year 700,000 to 1.2 million new cases....classified as neglected tropical diseases (NTDs)"to "Cutaneous leishmaniasis (CL) is the most common disease form caused by Leishmania parasite infection and considered a negletect tropical dissesse (NTD) affecting 700,00 to 1.2 million new cases per year in the world." Also include one phrase supporting the importance of metabolites studies to develop new drug targets.

Introduction: In lines 54,change the text to be clear:  1) infected female sandflies can transmite the parasite; b) promastigotes was regurgited during the bite into the skin; c) macrophages, neutrophyls and DC can phagocytize the parasite in the local; and d) promastigotes differentiate to amastigotes and after replicates.

Introduction: I miss the contextualization of glycerophosphocolines and lipid metabolism.

Methodology is clear and adequate.

Results: This section can be divided in to parts (subtitles) to explore the findings of PCs in the ear and in the footpad.

 I suggest to divide the Figure 1 in two figures, 1 - Figure 1A-D, increase the figure size (B, C and D) to better visualization of data; 2 - Fig E-M, to highlights the PCs.

In the tables:

- NA, Is the "not aplicable" term related to undetermined metabolite? Did you identifica the Number of Carbons and bounds?

  • Explain what is the means of the NA in the numbers of matched Peaks? Did you find the peak for each metabolites in all samples analyzed?
  • - The most significant data was found in the ear samples, I suggest to organize the table by metabolites dysregulated in the center and later in the edge compared to uninfected to highlights the important finds to reader. Because is confuse to visualize in the table what is the most important. Reduce the text inside the table. Create a  heatmap or circos figure.
  • -Can you use other tools to identify the undetermined metabolites? Do you tried the CEU Mass Mediator tool (http://ceumass.eps.uspceu.es)?

Did you find metabolites related to glutamine metabolism? What is the relation with PC metabolism?

The discussion section can be improved. Consider to explore that the local inflammation can recruit immune cells, other than macrophages, as T cells and DCs, impacting in the PCs available in the lesion. Its can be distinct in the ear and in footpad. The amount of parasite injected in the skin can interfere in the tem of cell recruitment and tissue recover. After 7-8 weeks of infection, what is the about of amastigotes in the lesion, because the parasite cells can impact in the PC collection.

Author Response

1. Title: I suggest to change the title "Dysregulation of Glycerophosphocholines in the experimental cutaneous lesion by Leishmania major in murine model".

Response: As recommended, we have modified the title, which now reads: “Dysregulation of glycerophosphocholines in the cutaneous lesion caused by Leishmania major in experimental murine models” (lines 2-3 in simple markup view and lines 2-6 in full markup view).

2. Abstract: I suggest to revise the first phrase "Each year 700,000 to 1.2 million new cases....classified as neglected tropical diseases (NTDs)"to "Cutaneous leishmaniasis (CL) is the most common disease form caused by Leishmania parasite infection and considered a negletect tropical dissesse (NTD) affecting 700,00 to 1.2 million new cases per year in the world." Also include one phrase supporting the importance of metabolites studies to develop new drug targets.

Response: we have made the recommended changes (lines 19-21 and 22-24 in simple markup view and lines 22-25 and 28-29 in full markup view). Phrase on metabolite studies for drug target identification now reads: “Studying disease metabolism at the site of infection can provide knowledge of new targets for host-targeted drug development” (lines 22-24 in simple markup view, lines 28-29 in full markup view).

3. Introduction: In lines 54,change the text to be clear:  1) infected female sandflies can transmite the parasite; b) promastigotes was regurgited during the bite into the skin; c) macrophages, neutrophyls and DC can phagocytize the parasite in the local; and d) promastigotes differentiate to amastigotes and after replicates.

Response: we have made the recommended change (lines 55-60 in simple markup view, lines 61-65 in full markup view). Sentence now reads: “Infected female sandflies bite the mammalian host, regurgitating promastigote-stage parasites into the host’s body. Subsequently, the parasites are taken up by macrophages, neutrophils and dendritic cells. Within the macrophage phagolysosome, promastigotes differentiate into the amastigote stage, which then multiply and affect various tissue types depending on whether infection is initiated by a viscerotropic or dermotropic parasite strain [4][5]”.

4. Introduction: I miss the contextualization of glycerophosphocolines and lipid metabolism.

Response: As recommended, we have added additional information on glycerophosphocholines in the introduction (lines 79-84 in simple markup view, lines 79-90 in full markup view).

5. Methodology is clear and adequate.

Response: we thank the reviewer for their appreciation of our approach.

6. Results: This section can be divided in to parts (subtitles) to explore the findings of PCs in the ear and in the footpad.

Response: we have made the recommended change. Results section is now divided into two subsections, one focused on overall changes (lines 89-166 in simple markup view and lines 96-187 in full markup view) and one focused on changes in PCs (lines 166-201 in simple markup view and lines 188-226 in full markup view).

 7. I suggest to divide the Figure 1 in two figures, 1 - Figure 1A-D, increase the figure size (B, C and D) to better visualization of data; 2 - Fig E-M, to highlights the PCs.

Response: we have made the recommended change. Figure 1 has been split into new figure 1 and new figure 4 (PCs only). 

8. In the tables:

NA, Is the "not aplicable" term related to undetermined metabolite? 

Response: Mass difference, cosine score and number of matched peaks all refer to the spectral matches on GNPS. Thus, this information is NA when we do not have a GNPS spectral match. We have clarified the table footnotes.

9. Did you identifica the Number of Carbons and bounds?

Response: As requested, we have added this information to the tables.

10. Explain what is the means of the NA in the numbers of matched Peaks? Did you find the peak for each metabolites in all samples analyzed?

Response: Number of matched peaks refers to the spectral match on GNPS. Thus, this information is NA when we do not have a GNPS spectral match. We have clarified the table headers accordingly.

 11. The most significant data was found in the ear samples, I suggest to organize the table by metabolites dysregulated in the center and later in the edge compared to uninfected to highlights the important finds to reader. Because is confuse to visualize in the table what is the most important. Reduce the text inside the table. 

Response: we have made the selected change in Table 1 and Table 2.

12. Create a  heatmap or circos figure.

Response: We have created the requested heatmaps (Figure 2).

13. Can you use other tools to identify the undetermined metabolites? Do you tried the CEU Mass Mediator tool (http://ceumass.eps.uspceu.es)?

Response: We thank the reviewer for their recommendation of this tool. All annotations are provided based on MS2 fragmentation spectra matched to library references or via molecular networking in GNPS, to enable greater confidence in annotations. All matches were manually inspected by comparing reference and experimental mirror plots, with only well-matching annotations retained. This ensures that readers only see high-quality validated annotations. We tried using CSI-Finger ID ( doi:10.1073/pnas.1509788112 ) to predict annotations based on MS2 fragmentation patterns, but this did not return any high-quality matches. All data is freely available, enabling readers to explore the data as desired or re-analysis as new tools continue to emerge. 

14. Did you find metabolites related to glutamine metabolism? 

Response: The only metabolite related to glutamine metabolism and highly affected by infection was glutamine itself. 

15. What is the relation with PC metabolism?

Response: We apologize for our lack of clarity. There is no relationship between our observations regarding glutamine and PC metabolism, except for the fact that they are both perturbed by infection. We have rephrased this sentence (lines 125-134 in simple markup view and lines 141-144 in full markup view).

16. The discussion section can be improved. Consider to explore that the local inflammation can recruit immune cells, other than macrophages, as T cells and DCs, impacting in the PCs available in the lesion. Its can be distinct in the ear and in footpad. 

Response: We thank the reviewer for this suggestion and have made the recommended edits (lines 285-292 in simple markup view and lines 273-276 in full markup view).

17. The amount of parasite injected in the skin can interfere in the tem of cell recruitment and tissue recover. After 7-8 weeks of infection, what is the about of amastigotes in the lesion, because the parasite cells can impact in the PC collection.

Response: Parasites persist in the lesion at the time of collection, as indicated by our bioluminescence data. We have provided an additional supplementary figure to demonstrate this (Figure S2). We cannot provide absolute parasite quantification since tissue material was used up for metabolomics and was thus not available for qPCR or limiting dilution. 

Round 2

Reviewer 3 Report

good